# Nondairy Probiotic Products: Functional Foods That Require More Attention

**DOI:** 10.3390/nu14040753

**Published:** 2022-02-10

**Authors:** Kübra Küçükgöz, Monika Trząskowska

**Affiliations:** Department of Food Gastronomy and Food Hygiene, Institute of Human Nutrition Sciences, Warsaw University of Life Sciences–SGGW, Nowoursynowska 159c St., 02-776 Warsaw, Poland; kubra_kucukgoz@sggw.edu.pl

**Keywords:** fermentation, functional food, nondairy food, plant foods, probiotic, in vitro digestion

## Abstract

The potential health benefits of probiotics have been illustrated by many studies. However, most functional foods containing probiotics are from dairy sources. This review provides an overview of potential strains and raw materials for nondairy probiotic products together with the role of its in vitro assessment. Probiotic-containing products from raw nondairy materials are known both in terms of quality and nutritional values. The sensory properties of raw plant-based materials are generally improved as a result of fermentation with probiotics. Increased market shares for plant-based probiotic products may also help to curb environmental challenges. The sustainability of this food results from reductions in land use, greenhouse gas emissions, and water use during production. Consuming nondairy probiotic food can be a personal step to contribute to climate change mitigation. Since some people cannot or do not want to eat dairy products, this creates a market gap in the supply of nutritious food. Therefore, the promotion and broader development of these foods are needed. Expanding our knowledge on how to best produce these functional foods and increasing our understanding of their in vivo behaviours are crucial. The latter may be efficiently achieved by utilizing available in vitro digestion systems that reliably recapitulate the in vivo situation without introducing any ethical concerns.

## 1. Introduction

Growing consumer interest in health and wellness also affects nutritional habits and food choices. Consumers’ nutritional understanding has changed from only meeting their energy needs to also providing a healthy and balanced nutrition profile. Functional foods including probiotic-containing products belong to this diet category [1]. Consumers are also becoming more concerned about the sustainability of the food chain; thus, this encourages manufacturers to give importance to the development of such functional foods. The key to the successful marketing and acceptance of new foods depends on the concept of added value based on food quality and food functions [2]. The global probiotic food market is growing very quickly due to increasing consumer awareness about the impact of food on health. Today, probiotic products account for 60% to 70% of the total functional food market [3,4].

Probiotics are a common ingredient in functional foods, as they confer health benefits when consumed in adequate amounts [5]. There are various health benefits associated with probiotic strains, including intestinal and nonintestinal effects. Intestinal benefits include the prevention of diarrhoea, the reduction in symptoms associated with inflammatory bowel disease, the prevention of gastrointestinal cancers, the alleviation of lactose intolerance, and a reduction in *Helicobacter pylori* infections [6]. Moreover, probiotics may play a role in the prevention and treatment of intestinal inflammatory disorders [7] such as Crohn’s disease and pouchitis, and paediatric atopic disorders. The impact of using probiotics on bacterial infections and immunological conditions such as adult asthma, cancer, diabetes, and arthritis is unconfirmed in humans [7,8].

The intestinal microbiota is as a potential factor in pathophysiology and associated metabolic disorders. Studies investigating the effect of probiotic intake on serum lipids, cholesterol levels, and more recently on blood pressure and glucose regulation indicate that probiotics may also benefit these factors [9].

Additionally, supplementing pregnant mothers with probiotics impacts mother and infant metabolism and later health [10,11]. The above-mentioned advantages of probiotics justify the indepth research of nondairy probiotic products, encompassing strain selection and characteristics, functional food development, and health properties.

The purpose of this review is to draw attention to and provide an overview of potential strains and raw materials for the production of nondairy probiotic products, along with the role of in vitro evaluation of such functional foods to accelerate the research and development of this functional food category.

## 2. Literature Search Methodology

For this review, a literature search was conducted in the Web of Science, PubMed, Google Scholar, and ScienceDirect search engines with keywords “fermentation“, “functional food”, “nondairy food”, “plant foods”, “probiotic”, and “in vitro digestion”. All selected terms were used in one search.

The timeline for our literature survey was set from 2011 to 2021 (in January 2022). The article titles and abstracts were reviewed, and duplicates were removed. Only studies on probiotics and nondairy food products were considered for inclusion. The literature concerning animals was excluded. Eligible sources of evidence included research articles, review articles, short communications, and book chapters. Full articles with appropriate references were obtained, the full text was read, and they were evaluated for final inclusion. Additional studies with respect to our search terms were only used for limited and specific purposes.

## 3. Potential Strains and Raw Materials for Nondairy Probiotics

### 3.1. Probiotic Strains and Viability Properties

The benefits of probiotic products are related to the selection of probiotic strains and their survival. The functionality of probiotics is generally strain-dependent. Strains should be resistant to gastric acid and bile, and be safe for human consumption [12,13]. Furthermore, a food must contain an adequate number of viable bacteria to have probiotic properties [14]. The stages of probiotic food production affect probiotic microorganisms’ viability and stability. Microorganisms should also survive during processing, storage, handling, transport, and shelf life [4].

Due to these criteria and safety regulations, *Lactobacillus*, *Streptococcus*, *Propionibacterium, Enterococcus, Pediococcus,* and *Saccharomyces* can be used as probiotic microorganism sources for nondairy probiotic products [15].

Fermented nondairy food products can also be a source of probiotic bacteria. For example, bacterial strains isolated from cucumber and cabbage prepared by traditional methods have probiotic properties. Ten different *Lactobacillus* strains were isolated: *L. johnsnonii* K4, *L. rhamnosus* K3, *L. brevis* (O22, O24), *L. plantarum* (O19, 020), and *L. casei* (O12, 013, 016, O18). Isolated strains were examined in gastrointestinal conditions to test safety for human consumption by in vitro experiment. Most of the isolated *Lactobacillus* strains could survive in gastrointestinal conditions and are safe for human consumption [12,13].

However, different raw materials play a specific role in bacterial growth, functionality, viability, and survival with their food matrix. Therefore, well-suited strains should be selected for each type of product [16,17,18,19,20]. Many studies have been conducted to incorporate microbes into different food matrices, some of which are discussed below.

Research has been undertaken to determine the suitability of tomato juice as a raw material for the production of probiotic juice by four lactic acid bacteria. Tomato juice was inoculated with probiotics such as *L. acidophilus*, *L. casei*, *L. delbrueckii*, and *L. plantarum*. The bacteria isolate fermented tomato juice from pH 4.1 to 3.5 in 72 h. They reached a viable cell population of more than 8 log CFU/mL (of <5 log CFU/mL) after 48 h of fermentation at 30 °C [18]. In other research, *L. sanfranciscensis* was added to tomato juice and stored for 4 weeks at 4 °C. After storage, the number of surviving bacteria was determined and there was a decrease in probiotic viability. However, decreasing amount from 8 to 7.5 log CFU/mL was still acceptable and showed that tomato juice is a possible carrier of probiotic *L. sanfranciscensis* [20].

Oats are important sources of beta-glucan, recognized as the most important functional component in cereal fibre. In addition, beta-glucan is known as a prebiotic as it stimulates the growth of some beneficial microorganisms in the colon [21,22]. Furthermore, beta-glucan supports the viability of probiotic strains during cold storage [21,22,23]. In a study, the effects of beta-glucan obtained from oatmeal and modified beta-glucan samples obtained with xylanase treatment on the probiotic *Bifidobacterium bifidum* were investigated. While the two components had a significant effect on the growth of *Bifidobacterium bifidum*, the effect of modified beta-glucan was greater [22,23,24].

Beetroots have rich nutrient content and bioactive compounds [25,26]. Fermented beetroot with *Lactobacillus* bacteria had good biological viability and antimutagenic activity for up to 30 days at refrigerated storage [26]. Research about enriched ready-to-eat beetroot products with *L. plantarum* showed 8–9 log probiotic cells in 100 g. In addition, probiotic viability was greater than 7 log CFU/g after 21 days of storage at 4 °C; these results showed that the beetroot food matrix is favourable for probiotic survival [27].

Research that focused on producing potentially probiotic orange juice showed that different microorganisms have different viability. *L. rhamnosus* and nettle (*Urtica dioica* L.) additions were used for production, while *L. rhamnosus* was able to remain above 6 log CFU/mL at 4 °C storage for 28 days, but nettle could not improve the viability of the product [28].

Another study reported on orange juice with *Bacillus coagulans* GBI-30 6086 in animal models compared with yoghurt samples of the same probiotic. The probiotic orange juice food matrix adversely affected the functionality of probiotics in rats. The rats that had been fed the probiotic yoghurt group also showed higher gut bacterial diversity than that of orange juice [29].

Some features of the raw materials can cause the loss of viability of probiotic microorganisms such as natural antimicrobial compounds, acidity, diacetyl, and hydrogen peroxide [30].

Table 1 provides data on probiotic bacterial viability in different types of food.

### 3.2. Properties and Environmental Concerns of Raw Nondairy Materials for Probiotic Products

All over the world, most probiotic products are dairy-based. The increased health awareness of consumers and some health-related issues has led to the exploration of nondairy-based products. For example, plant-based alternative yoghurts are being developed and marketed in increasing numbers [40]. Statistical analysis shows that there are more than 380 types of probiotic products in the world, but 80% of these products are from dairy sources. Nondairy probiotic products with fruit and vegetable origins are very rare [5]. The lack of nondairy probiotic products means that various human groups do not benefit from functional foods containing probiotics. However, industry and people’s interest in nondairy probiotic products is increasing for a variety of reasons [41]. The strongest drivers of nondairy products are vegetarianism, milk cholesterol content, lactose intolerance, and consumer interest in shelf diversity and sensory appeal. From the industry viewpoint, many manufacturers are seeking ways to create and increase value, which has further increased the product profile. However, a more compelling reason and the stronger driver is the emerging evidence of health benefits that can be acquired from a symbiotic relationship between plant components and probiotics, and gut commensals [42,43]. Nondairy products also contain more antioxidant phytochemicals such as phenolic acids, carotenoids, and flavonoids that have positive effects on oxidative stress in the body, prevent cell damage, and help to change the lipid metabolism and reduce obesity risk factors [43].

Considering product categories, cereal- and legume-based products increase their nutritional quality by a fermentation process using lactic acid bacteria and probiotic microorganisms [44]. Cereals have a rich content of dietary fibre, carbohydrates, and vitamins. Their nondigestible carbohydrate content also helps the growth of probiotic microorganisms in the human colon such as *Lactobacilli* and *Bifidobacteria*. Microbial processes on cereals such as fermentation also affect the improvement of protein digestibility and the reduction in allergens with microbial proteases [45,46]. Moreover, water kefir increased beneficial short-chain fatty acid production at the microbial level, reduced detrimental proteolytic fermentation compounds, and increased *Bifidobacterium* genus abundance [47]. Vegetables and fruits are also used in the production of nondairy probiotic products. These products have excellent nutritional values due to the presence of many phytochemicals, antioxidants, zero cholesterol, vitamins, minerals, and dietary fibre [1]. Fruit and vegetable juices can improve the viability of probiotics because additional nutrients can be obtained from the raw material by cellular synthesis, which is similar to the processes used during the fermentation of fruit and vegetable juice. This may make them ideal substrates for probiotic growth. Cutting or grating vegetables and fruits also helps to release their cellular content of vitamins, minerals, sugars, and other nutrients, and creates a good environment for probiotic microbial growth [31,48].

It is possible to find nondairy probiotic products in the market with different combinations of food matrices [18,26,49]. Nondairy probiotic beverages, frozen desserts, spoonable products, and probiotic vegan milk replacements are already on the market. Nondairy probiotics and prebiotics also have a great marketing future, as recent research shows the application of strains that are well-suited to alternative matrices [42,50,51].

In recent years, plant-based dairy alternatives have received more attention due to consumer demands and environmental concerns [52]. The production of dairy products is related to environmental externalities, including greenhouse gas emissions, soil degradation from overgrazing, soil erosion, deforestation, loss of biodiversity, the contamination of surfaces and groundwater arising from waste management, and soil salinization [53]. The report of the Lancet Commission on Food, Planet, and Health explained that contemporary research concluded that vegetarian and vegan diets are associated with reductions in land use, greenhouse gas emissions, and water use [54].

There is, therefore, another motivation to develop and popularize nondairy and vegan products that are involved in reducing climate change to encourage personal actions to reduce individual carbon footprints by switching to a plant-based diet. This kind of diet also helps to prevent diet-related chronic diseases and decrease expenses [45,55,56].

This growing interest in plant-based diets not only impacts sustainable consumption behaviour, but is also being noticed by the food industry [57]. Companies in the food sector need to create innovative products on market research while developing marketing skills in addition to scientific and R&D capacity [2]. Reasonable prices and lactose-free content increase the demand for these products.

Short-term marketing strategies should focus not only on vegan consumers but also on consumers who want to reduce their consumption of animal products and are looking for new strains of nonanimal origin.

### 3.3. Sensory Properties

Acceptable sensory properties are most important in probiotic food production, and are directly related to product quality, consumer acceptability, and processing characteristics [18,46]. Sensory changes can occur while producing probiotic products after the probiotic bacteria had been added to raw materials. Probiotic microorganisms produce different metabolic compounds such as lactic acid during storage and fermentation. Probiotic microorganisms also ferment the raw materials’ carbohydrate content, and increase alcohol content and production gases. This also affects the consumer acceptance of the product [58]. The development of nondairy probiotic products with vegetables and fruits can be undertaken in three different ways, namely, the fermentation, nonfermentation, and minimal processing of raw materials. Probiotic cultures and fermentation can also affect sensory aspects. For instance, lactic acid fermentation of fruits and vegetables enhances sensory and nutritional quality, and retains nutrients and coloured pigments [31,59]. Probiotic blackcurrant juice prepared with *L. plantarum* strains and blackcurrant juice have more acceptable sensory properties to consumers, such as flavour, appearance, aroma, and texture [58]. Another study regarding the fermentation of grape juice found that the sensory properties of a probiotic product prepared with *Lactobacillus rhamnosus* strains were highly regarded by the consumers [60].

Raw cereal grains do not have enough active organoleptic compounds with their taste and texture. This situation also affects the preferences of consumers. Fermentation can lead to reducing flavouring additives to cereals. In particular, lactic acid bacteria’s enzymatic activity on cereals contributes to the taste changes, such as the sweet and sour taste generated from nonvolatile and volatile compounds [60].

Plant-based milk is one of the most common materials used to produce probiotic beverages. It has a similar appearance to animal milk but offers different sensory properties, kinetic stability, and nutrient composition. In general, plant-based milk substitutes can be defined as homogenised extracts of vegetable matrices such as cereals, vegetables, and nuts. The nutritional profile of plant-based milk alternatives is usually unbalanced, and their flavour profiles limit their acceptance. Probiotic fermentation was shown in several studies to improve sensory acceptability compared with unfermented alternatives [61,62,63].

Many researchers proved and studied that probiotic cultures did not affect the overall acceptability of the products, but these products are yet to come to the market [61,63].

## 4. In Vitro Assessment of Probiotic Product by Artificial Gastrointestinal Tract

The health-promoting effects of probiotics often depend on their survival during transit through the gastrointestinal tract. To show health benefits, probiotic microorganisms should be resistant to digestion conditions and colonise in adequate amounts in the host [64,65]. Their survival rate depends on some factors such as galenic form, food matrix, and dosage. To prove and understand the beneficial effects and the survival of probiotic microorganisms in the host, the passage of these microorganisms must be observed throughout the gastrointestinal transit [66]. However, it is difficult to investigate this phenomenon with in vivo study. Research shows that in vitro models of the upper and lower gastrointestinal tract can provide significant insight into the behaviour of probiotic strains during digestion in humans. They are particularly relevant for screening purposes, such as for studying the effects of biopharmaceutical factors (such as dosage form, food matrix, and dose regimen) on the viability of probiotic strains throughout the human digestive tract [67].

In addition, in vivo studies can be complex and expensive to investigate microorganisms. On the other hand, in vivo probiotic research generally focuses on the recovery of beneficial microorganisms from faeces, which makes it difficult to observe probiotics’ behaviour on the gastrointestinal transit. All these reasons show the importance of in vitro research in probiotic studies. Artificial digestion models are also quicker, less difficult to undertake, and have fewer ethical concerns. Research tools of artificial digestion tracts help in understanding chemical and structural changes of food components in the digestion tract parts and the gut microbiome [67].

Two different in vitro digestion models are developed, namely, static and dynamic digestion models, and they are used for research purposes. Generally, protocols for static digestion systems describe food in bioreactors where enzymatic, physical, and chemical conditions of each digestive part are recreated. However, these digestive models have limitations because digestion is a dynamic procedure. In these systems, there is no possibility to replace food between the different digestive parts, and environmental conditions such as enzymes, bile concentrations, and pH are stable [68].

Dynamic digestion models have better simulation advantages, such as physical conditions with constant biological and chemical changes. Generally, dynamic digestion models mimic all sections of the gastrointestinal tract for complete simulation. The main difference between dynamic digestion systems is configuration. Currently, the TNO artificial gastrointestinal model with specific variations (TIM-agc, tinyTIM, TIM-1, TIM2) is used [62]. The mainly used generic platform is TIM-1, which includes the stomach, duodenum, jejunum, and ileum. These four compartments are connected with peristaltic valve pumps. This configuration has several variants for animals and humans for different kinds of meals. TinyTIM does not include separate intestinal steps and is a more basic version of TIM. TIM-agc is a more qualified version of TIM systems and it helps to compare the compounds of digestion under controlled conditions. As it is possible to observe the movement of foods and drugs, the design of this version enables a more accurate assessment of the behaviour of the stomach [57].

Another currently used dynamic digestive system is the Simulator of Human Intestinal Microbial Ecosystem (SHIME^®^) model, which is a computer-controlled gastrointestinal simulation device. It is possible to examine the microbial ecology and physiology of the gastrointestinal system. The model allows for simulating various age groups and some diseases. The simulator consists of five different reactors that help to see parts of the gastrointestinal system, the stomach, the ileum, and three parts of the colon (ascendant, transversal, and descendent). First, reactors allow for simulating steps of food intake and digestion with fill and drawing reactor. The peristaltic pumps, SHIME^®^ nutritional medium, pancreatic enzymes, and bile liquid set off physiological conditions in the large intestine [69,70]. The model also maintains microbiota stability for a determined time and helps to observe the adaptation of microbiota. The different subjects can also be examined at the same time and the subject’s microbiome can be stored to set up unique features. The Mucus SHIME^®^ is the specific variation of the SHIME^®^ model. The model is used for the investigation of the adhesion ability of bacteria and changes in the microbiome in the mucosal parts of the gastrointestinal system [71].

For instance, the following studies were conducted on in vitro digestion models. *Lactobacillus crispatus* strain, added to cheese as a probiotic culture and isolated from a healthy human vaginal environment, was tested for its digestion system process using SHIME^®^. Results showed that the survival of *L. cripatus* BC4 was not affected by gastric digestion, but was significantly affected by bile salts and pancreatic juice. During colon simulation, *L. cripatus* BC4 was able to grow under sterile colon conditions and survive in the presence of a complex microbiota [72]. Another study also investigated soybean polysaccharides’ bioavailability and the metabolites on the gut microbiota by using SHIME. Results showed that soybean polysaccharides were only partially decreased in the oral, gastric, and small intestine parts of SHIME [73].

Increasing our understanding of probiotic behaviours in the product and during gastrointestinal passage is crucial in the development of nondairy probiotic food. This may be efficiently achieved by utilising available in vitro digestion systems that reliably recapitulate the in vivo situation without introducing any ethical concerns.

## 5. Conclusions

The potential health benefits of probiotics have been illustrated by many studies. Most of the functional foods containing those beneficial microorganisms are from dairy sources. However, the high fat, cholesterol, lactose, or allergen content of dairy products may induce health problems and cause the exclusion of valuable functional foods from the diet. One of the solutions to this problem may be products containing probiotics produced from nondairy raw materials. The value and benefit of the probiotics themselves, combined with raw plant materials, give rise to unique advantages, for example, additional content of fibre or phytochemicals with quality sensory properties.

There is a market gap in the supply of the discussed nutritious food, especially for people who are unable or unwilling to eat dairy products. To address this issue, there is a need to intensify the indepth research and development of nondairy probiotic foods. In particular, advances in product evaluation through in vitro digestion models lead to faster and more accurate data on the health value of the product. In vitro artificial digestion systems are reliable, and this research methodology has no ethical concerns.

In addition, paying attention to nondairy and vegan foods benefits the environment by reducing land use, greenhouse gas emissions, and water consumption compared to the production of raw dairy materials.

## Figures and Tables

**Table 1 nutrients-14-00753-t001:** Viability of probiotic bacteria in the different types of foods.

Genus	Species	Product Type	Viability (log CFU per mL or g)	References
*Lactobacillus*	*L. rhamnosus* ATCC7469	Fruit-Based Product Dried apple slices	1.0–3.0 log in slices dried by freezing and a combination of air drying and vacuum drying after 120 days storage at 25 °C, but higher viability of 9.3–7.8 log was found at 4 °C for 180 days.	[31]
	*L. plantarum* B2, *L. fermentum* PBCC11.	Fruit-Based Product Fresh-cut cantaloupe	*L. plantarum* (8.1 log) and *L. fermentum* (7.8 log) after 11 days of storage at 4 °C	[32]
	*L. helveticus* 76 (Lh76)	Fruit-Based Product Kiwifruit juice	Above 9.0 log CFU/mL after fermentation	[33]
	*L.delbrueckii* subsp. *bulgaricus*	Legume Based ProductSoy Protein	First day after fermentation 54 × 10⁶ CFU/mL, after period of 15 days 43 × 10^7^ CFU/mL	[34]
	*L. paracasei* LBC-81	Cereal-Based Product Maize-based substrate	Viable cell count, 10^6^ CFU/mL	[35]
	*L. reuteri* NCIMB11951	Grain-based Product Fermented beverage made from oats, barley or malt	Viability between 7.8 and 8.1 log of the three species in fermented beverage after 10 h of fermentation at 37 °C.	[36]
	*L. johnsonii*	Vegetable-Based Product Traditional fermented cabbage and cucumber	Above 9 log CFU/g	[12,13]
	*B. bifidum*	Fruit-Based Product Blueberry and Black Berry Juices	Increased CFU/mL and 7.3 log_10_ CFU/mL to 8.2 log_10_ CFU/mL after 48 h fermentation,	[37]
*Bifidobacterium strains*	*B. lactis* Bb-12	Fruit-Based Product Cashew apple juice	After 1 day fermentation 2.16 × 10^10^ CFU/L h	[38]
	*B. longum* *Bifidobacterium longum* Bb-46	Fruit-Based Product Apricot Fruit Juice	After 24 h of fermentation were higher than 10^8^ CFU/mL,	[38]
*Saccharomyces*	*Saccharomyces cerevisiae* CCMA 0731,	Cereal-Based Product Maize-based substrate	Viable cell counts 10^6^ CFU/mL	[35]
*Streptococcus*	*Streptococcus thermophiles*	Grain-based product Oat Flour	Viable cell counts 10^6^ CFU/mL	[39]

## Data Availability

Not applicable.

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
