# Peer review of "Nondairy Probiotic Products: Functional Foods That Require More Attention"

_nutrients, 2022, doi:10.3390/nu14040753_

Round 1

Reviewer 1 Report

Introductory remarks

This review considers the development of functional foods containing probiotic bacteria and specifically advocates the need for developing non-dairy products within this product line. The authors point out that most probiotic containing foods available today are based on dairy products, which may prevent consumers from using them, due to ideological reasons (eg veganism or environmental impact) or food allergies. Hence, they make a case of the necessity to develop probiotic functional foods on basis of non-dairy raw material, for example vegetables, legumes, and fruits. The authors further emphasize that it is essential to examine probiotic candidates by in vitro systems, to understand their behavior and potential beneficial effects throughout the gastrointestinal canal.

The manuscript put forward some interesting perspectives, in a rather “hot” area. The interest of functional foods is shared by the academia, industry, health care and public, which makes it an interesting topic to address. 

General Concept Comments

The objective of this review was to provide an overview of potential probiotic strains and raw materials for non-dairy probiotic products, together with the role of an in vitro assessment of such emerging functional foods. The authors also set out to discuss the attitudes of consumers towards this type of products and their market potential.

The introductory section makes case for a conceived need of this review. However, this section could be more stringent, and the language needs to be considerably improved.   

1. I advise, a proper language check of the entire manuscript, either from a native English speaking fellow scientist or from a professional providing this service.

2. Please also consider my comments on this section, as regard to references and the stringency, in my Specific Comments following below.

I am deliberately writing “conceived need” of this review, above, because I am not entirely convinced on the actual need for it. Some of the “key” papers referred to are almost over (or over) a decade old (see for example ref 60 and 61, or 66 and 67) which makes the novelty, and therefore also the need, of this review questionable. This is also my personal opinion as a researcher within this field; the idea of plant-based functional foods containing probiotic bacteria is not new. But with that said, the review touch on an interesting topic and could potentially be make a case to push method development and industry to move faster within this area. 

3. I suggest that the review is re-written/re-structure to highlight what novel concepts or insights it brings to the field. Probiotic, vegetable-based functional foods are not new enough as many are already on the market. However, the need for promoting faster and broader development of these foods and putting forward systematic ways of evaluating probiotic candidates, suitable “raw-materials” and consumer acceptance may be important to push for.

The way the review is presented in its current form, it comes across as “narrative” (rather than systematic) in its approach (ie it arbitrarily identifies a set of studies that describe a problem of interest, without a specified search strategy). If this is NOT the case, the authors need to incorporate a thorough methods description, as it is completely lacking at this stage.

4. I kindly ask the authors to clarify the methodological approach in this review.

4a. IF this review has been composed through any structured method, I strongly advice the authors to include a Method Section in the manuscript as this would strengthen the quality.

Specific Comments 

Below I comment on specific parts in the manuscript and suggest how to improve them accordingly. A general note that I want to stress, is that all parts needs further language editing.

Abstract:

I would question the use of the word “proven” in line 9, as this can be debated. Most systematic reviews on probiotics and their benefits are inconclusive.

5. I suggest that the first sentence of the abstract is reformulated to better illustrate the true state of this field and how much benefit probiotics actually can account for.

The authors need to make a clearer case for the need, and rational, of this review.

6. I suggest that the authors stress the information stated in line 136-138 (ref no 3). Namely, that the market suffers from an overweight in dairy-based products and that this is a problem, because of consumer needs and wants and environmental challenges, which needs to be urgently addressed.

7. The abstract would also benefit from a clear objective and a conclusion/future perspective that aligns better with the content of the manuscript. Please incorporate this (also see comment 12, 27 and 28 touching on these parts for the main document).

Introduction:

In general, the Introduction needs to be “streamlined” and more stringent, ie making a case for the review and what is to come, by (only) introducing key concepts and information of importance to the review rational. For example,

8. Improve the sentence in line 31-33 and integrate it with the section below, starting at line 34.

9. Revise the sentence in line 37-38 to better correspond to the clinical use of probiotics and include appropriate references. Probiotics in prevention and treatment of the chronic inflammatory conditions referred to are hypothetically interesting but seldom used in clinical care. So far, their effectiveness is questionable (on basis of current studies) and the evidence is not strong enough to advice on evidence based clinical practice.

10. Rewrite the section found between line 42-45. Reference no 6 is an excellent paper but here it is referred to somewhat outside of the scope of the current review, as this section (line 42-45) comments on the microbiota changes in different metabolic conditions and not on the use of probiotics.

11. The section found between line 46-50 falls outside of the scope of the review as it is currently written. The use of probiotic in this very specific patient sub-group is indeed interesting, but it should be mentioned very briefly (preferably in one sentence informing on the hypothetical benefits of probiotics in different medical conditions, incorporating multiple references) - if mentioned at all - as it pulls the reader from the essence of the review.

12. Line 51-53 reads very well, but make sure that the objective that follows corresponds to what is the actual essence of the review. From the way it is currently written it promises more than what is delivered on. To address this the objective could either be rewritten to better correspond to the actual content of the paper OR the “deliverables” should be better highlighted/presented in the paper. For example, delivering on the purposed “overview of potential strains” by presenting them in a structured table (including references) and a similar table for presenting potential “raw material” (see comment 14 and 15 below).

Text body:

Potential strains and raw materials for non-dairy probiotic

Probiotic Strains and Viability Properties

13. Rewrite the sentence between line 61-64 and simultaneously consider that probiotic bacteria can confer health benefits to the host without “adhering”. However, if you supplement with a bacterium that does not colonize (which many probiotics fail to do as they cannot be detected in stool after the supplementation stops) you need to consider continuous treatment.

14. The probiotics mentioned in the section between line 69- 78 could be presented in a table. However, the reader needs to be much better informed on the advocation of these particular strains in the text proceeding this section/the table. This is a proposed “core deliverable” in this review, as the objective is currently stated, and the case for suggesting these exact bacteria/strains needs to be crystal clear.

15. The next proposed deliverable is the potential raw materials to be used. The sentence at line 90-91 states that “many studies have been done /.../ some of which are discussed below”, if this is a core deliverable in this review this section should bring up most (if not all) of these studies. They should also be condensed and presented in a table with corresponding references. (Here the review could have greatly benefited from a more systematic approach in retrieving appropriate and contemporary literature to cover the key publications in this field).

16. The information on line 127-129 is important for this paper. However, Table 1 tells nothing about how thorough the literature review of this particular information has been. Are these all of the available contemporary studies considering probiotic viability in this context or is this an arbitrary selection? What was considered when incorporating this exact set of studies? Are there more to be found (I strongly sense that there are) and if so, why are these not incorporated?

Properties of Non-Dairy raw materials for Probiotic Products

17. The information conveyed in line 151-154 is questionable in its accuracy, particular on basis of the wording “/…/ that have positive effects on removing oxidative stress from the body /…/”. I argue that oxidative stress cannot be removed from the body and further that is a very crucial biological response in many “normal” life-sustaining physiological processes, for example in skeletal muscle adaptation and immune responses.

18. The section following through line 163-175 brings in an interesting perspective of importance to this review and should be expanded on. However, it would fit better in adjacence to the section on In Vitro Assessment Of Probiotic Product By Artificial Gastrointestinal Tract.

Food Matrix Properties for Probiotic Strain Adaptation

19. The reference in line 190 is not properly formatted.

20. Clarify the meaning of the sentence starting in line 191. I also suppose that this should not be its on paragraph; please merge it with the above sentence. Also, on that note, take care to divide the text into reasonably large sections throughout the manuscript (one, or a couple of sentences does not make an entire paragraph).

21. Please include proper references to the information through line 194-198.

22. Please better specify the cereals referred to in line 204. Not all (I would say quite few) cereals have a rich content of protein.

Sensory properties

23. This is an important section of the review that would benefit from being highlighted and enriched with some more information and reasoning.

In Vitro Assessment Of Probiotic Product By Artificial Gastrointestinal Tract

24. This is an important, and rather well written part, of the review. It would however benefit from some enrichment. I would also like to suggest that the authors complement this part with a figure, depicting the in vitro system/s they propose.

Probiotic-Based Non-Dairy Innovative Foods As a Future Functional Foods and Market Trends

Much of the information in this section could be introduced earlier and would benefit from integration with previous parts to avoid repetition.

25. In line 297 the authors use the term “recent” to describe research that is everything else but recent. Please, rephrase.

26. Also in line 300, development of non-diary probiotic products in referred to as “future”. I would argue that it - the proposed future - is already here. Please, rephrase.

27. My advice is to take out this whole section and integrate it with the other parts. I think it would elevate the review to finish off with the in vitro assessments. By doing so I also think that the author will have an easier task in writing a thought through conclusion.

Conclusion

28. The conclusion should be rewritten to correspond to the changes suggested above. Make sure to clearly convey the take-home message of the review, as this may well be said to be its most important part; especially in the form of a narrative review.

Reviewer 2 Report

The  review was well writtent but the plagiarism was 28%, please reduce the similarity in the text   

Reviewer 3 Report

This manuscript is very well written. However, some comments should be adressed before their publication in Nutrients.

  • Please use the new taxonomy for most bacteria described in the manuscript. For example, Lactobacillus casei is called now as Lacticaseibacillus casei, Lactobacillus salivarius as Ligilactobacillus salivarius, Lactobacillus plantarum as Lactiplantibacillus plantarum, Lactobacillus fermentum as Limosolactobacillus fermentum.
  • Please include some figure that represent the aim of the manuscript
  • Please double chechk the references section. For example the abreviation of the Journal name is not the homogeneous for the references 55 and 60, 24 and 27. The reference 23 have a different format than the rest. 

Round 2

Reviewer 1 Report

Abstract

  1. Take another look at this sentence, line 16-17:

Plant based non-dairy products may equally help the curb enviromental concerns of consumers which needs to be urgently addressed.

To me it is not reading well, as it tries to fit in too much information. For one, increased market shares for plant-based probiotic products may help to curb environmental challenges, and yes, current global environmental challenges need to be addressed urgently. This, the fact that choosing plant-based probiotic products over dairy ones may have positive environmental effects, could be interest to consumers that are increasingly aware of how their food choices impacts the environment. Please rephrase or state this information in two separate sentences.

Also, the word environmental is misspelled.

  1. After improving the point above, also take a look at the text through line 17-21. This could now be restructured to follow in a more logic order.

  1. I would suggest rephrasing the last sentences:

“This entails researching and understanding what happens to the probiotics in the new food matrix during intestinal transit. In order to observe the behavior of probiotics in these food matrices, in vivo systems are used that provide faster results, simpler ethical conditions and more reliability. “

For me this would, for example, read better accordingly:

Expanding our knowledge on how to best produce these functional foods and increasing our understanding on their in vivo behaviors are crucial. The latter may be efficiently done by utilizing available in vitro digestion systems that reliably recapitulate the in vivo situation without introducing any ethical concerns.

  1. Line 32 would read better if “what is more” is replaced by “Also” or another conjunction that the authors feel is appropriate.

  1. I would add “development” to the sentence line 33-34:

/…/ encourages manufacturers to give importance to the development and promotion”

As this review is pushing for new developmental approaches rather than promotion (although the latter is an important aspect of industrial activities to increase market shares and hence is also reasonable to mention)

  1. The sentence in line 36-37 needs improving:

“Probiotics have been used for functional food products because adequate amounts can improve health as a microbial dietary supplement”

It would for example read better

“Probiotics are a common ingredient in functional foods, as they confer health benefits when consumed in adequate amounts.”

  1. The text in line 37-40, would fit great as the very initial piece of the Introduction as it “sets the stage” in a very distinct way. Moving this part up would also avoid repetition of similar information stated earlier in the introduction. If moved, take care to format the following text to maintain a good flow without repetition of information.

  1. The sentence in line 41-42 is somewhat ambiguously worded:

“There are various health benefits associated with probiotic strains, ranging from intestinal to non-intestinal problems.”

Benefits clashes with problems and needs to be replaced. I would suggest:

“There are various health benefits associated with probiotic strains, ranging including from both intestinal to non-intestinal problems effects.”

  1. In line 42-45 the authors draw from a review published in Engineering to claim that probiotics can have preventative effects on a number of ailments including GI cancers. If the authors want to make this claim (that are somewhat controversial and still poorly supported in contemporary scientific literature) they need to trace this information back to the original articles as cited in the review to provide a more nuanced picture of their claim. I believe that the authors of the cited review also have a similar problem, in making claims about probiotics and their preventative effects on, for example, GI cancers that over exceeds the findings of the original research they are referring to…

Please make sure that his kind of referring, ie not referring to original research or referring to breif remarks in reviews, does not occur elsewhere in the manuscript. All information should be traced back to the original source. IF the cited review is not a high-quality systematic review/meta-analysis that can draw its own conclusions on firm empirical grounds.

  1. I would insert a may in the following sentence, line 46-48.

Moreover, probiotics may play a role in the prevention and treatment of intestinal inflame /…/

And, add a reference to this statement.

  1. Line 48. Remove: The gut microbiota promotes host health and well-being.

  1. The sentence in Line 48-50 needs improving.

The impact of using pro-biotics over bacterial infections and immunological conditions like adult asthma, cancer, diabetes, and arthritis is yet to be proven in humans.

  1. Line 51. Change the order in: “The intestinal microbiota has also been been also identified as a potential factor in pathophys/…/

  1. Between line 51-59 there is several short paragraphs of two (or even one) sentences. Please combine these in to one paragraph.

Literature and search methodology

Thanks for including and clarifying this part.

  1. Line 66-68. Did you include all these search terms in one search or was there multiple searches done? Pleas clarify this.

  1. The text in this section needs editing and (again) it contains very short paragraphs. It could for example look like this:

The timeline for our literature survey was set from 2011 to 2021 and the search was performed in January 2022. Additional literature with respect to our search terms were only used for limited and specific purposes. The article titles and abstracts were reviewed, and duplicates were removed. Only studies on probiotics and non-dairy food products were considered for inclusion. Consequently, from the search results Literature concerning animals was excluded. Eligible sources of evidence included research articles, reviews articles, short communications and book chapters. Full articles wit§h appropriate references were obtained read in full text and validated evaluated for final inclusion. Additional literature, with respect to our search terms, were only used for limited and specific purposes.

Potential strains and raw materials for non-dairy probiotic

Probiotic Strains and Viability Properties

  1. Short paragraphs, please merge.

  1. Line 89. Replace product with products

  1. References are missing in Tabel 1. Including references providing grounds for the suggestions/usage of these exact organisms are needed.

  1. Line 98. Change the wording:

“/…/ to teste its test their safety for human consumption by in vitro experiment”

  1. Line 101. Change the wording:

“/…/ raw materials have a specific role on in bacterial strains growth /…/”

  1. Line 101-105. Short paragraphs, please merge.

  1. Line 106. “Research” is repeated, remove the second.

  1. Line 118. I agree, beta-glucan are potent prebiotic fibers, but please add a reference to this statement.

  1. Line 120. Merge this section with the paragraph above, i.e. just move it up so it is no longer a new paragraph but rather a continuation of the beta-glucan introductory paragraph.

  1. Line 139-140. Change wording to clarify:

“Besides, feeding with the the rats fed probiotic yoghurt showed a higher gut bacterial diversity as compared to the group fed with orange juice”

Properties and environmental concerns of Non-Dairy raw materials for Probiotic Products

  1. Line 150-180. Many short paragraphs, please merge.

  1. Line 187-188. Cahnge the wording to clarify:

“/…/ many researchers have assessed contemporary research have concluded that vegetarian and vegan diets are associated with reductions in land use greenhouse gas emissions and reductions in water use.”

  1. General comments to improve this section:
    1. Take care to merge related pieces of text to avoid many short paragraphs.
    2. Work through this text again and take care to remove any repetitions. This could be done by restructuring the text so everything relating to the same subject is addressed in the same paragraph. For example, merging all information on “Vegetable, fruit, cereals and legumes” into one informative paragraph.

Food Matrix Properties for Probiotic Strain Adaptation

  1. Line 211. Do not start a new section with a conjunctive adverb like Therefore. It aims to connect the previous information with what is to follow in a way that is not appropriate after a newly introduced heading.

  1. Line 217-220. Information about free radicals is repeated, without reference. I would suggest this information is removed from this section to avoid repetition of information or re-written with included references.

  1. General comment: This section needs to be worked through to create a better flow and to justify its “being”. As it is written now it is not clear what it brings to the review. Some information is repeated, and it appears somewhat disorganized. Maybe it can be removed altogether? If some information is “precious” it can be moved to other parts of the manuscript. Perhaps to the following section (bleow), accompanied by a small change in the subheading to better reflect the content?

Sensory Properties

  1. The whole section contains many short paragraphs, please merge related text to avoid too short units.

  1. Line 267. Reference is missing.

In Vitro Assessment Of Probiotic Product By Artificial Gastrointestinal Tract

  1. Line 271. I would argue that the probiotics referred to here colonies in the host and not on the host. Please rephrase.

  1. Line 272-276. The two sentences repeats the same information, please merge.

  1. Line 288. Replace “conditions” with “concerns”.

  1. Line 291. The models have already been developed right? Replace “are” with “have been”. And, I believe you are referring to “static” and “dynamic” systems here. Please clarify by stating this at the end of the sentence.

  1. Line 300. Re-word:

“/…/ use mimics all the sections of the gastrointestinal tract for complete stimulation simulation.”

  1. Line 302. Is mainly used? I think adding mainly I would improve this sentence.

  1. Line 321. Merge this short paragraph with the previous.

  1. Line 324. Merge this short paragraph with the previous.

  1. Line 330. Use the abbreviation SHIME as it has already been introduced.

  1. Line 335. Use the abbreviation SHIME as it has already been introduced.

  1. Line 338. Please end this long section with a brief take-home-message to bridge over to the Conclusion.

Conclusion

  1. Line 344. Repetition of “However” (it appears also in the beginning of the previous sentence). Please correct.

  1. Line 345. “Cause” is repeated in the same sentence. Please correct.

  1. Line 347. Add a comma (,) after “land-use”.

  1. Line 348. No need to write “reductions in” water use as this is already stated earlier in the sentence.

  1. Line 349. Plant-based probiotic foods are good but I don’t believe that they are the “solution to this situation” (especially as the use of these words in this context points towards the “situation” to be global climate change), please rephrase.

  1. Line 358-359. I suggest you remove the last scentence.

Reviewer 3 Report

The manuscript has improved substantially and most of the reviewers' questions and concerns have been addressed.

Author Response

Dear Reviewer,

Thank you for your comment.

Kind regards,

Kübra Küçükgöz and Monika TrzÄ…skowska